# Effects of Different Highland Barley Varieties on Quality and Digestibility of Noodles

**DOI:** 10.3390/foods14132163

**Published:** 2025-06-20

**Authors:** Guiyun Wu, Lili Wang, Xueqing Wang, Bin Dang, Wengang Zhang, Jingjing Yang, Lang Jia, Jinbian Wei, Zhihui Han, Xiaopei Chen, Jingfeng Li, Xijuan Yang, Fengzhong Wang

**Affiliations:** 1College of Agriculture and Animal Husbandry, Qinghai University, Xining 810016, China; 19535802557@163.com (G.W.); 2008990019@qhu.edu.cn (B.D.); 2017990098@qhu.edu.cn (W.Z.); 2Institute of Food Science and Technology, Chinese Academy of Agricultural Sciences, Beijing 100193, China; wlland2013@163.com (L.W.); wxqing199701@163.com (X.W.); yjj4676@163.com (J.Y.); 15551299084@163.com (J.L.); 3College of Food Science and Engineering, Tianjin University of Science and Technology, Tianjin 300457, China; jl15135408007@163.com (L.J.); weijinbian202403@163.com (J.W.); zhihuih261@gmail.com (Z.H.); 13103302240@163.com (X.C.)

**Keywords:** highland barley varieties, noodle quality, digestive characteristics

## Abstract

This study comprehensively assessed the effects of ten highland barley varieties on the quality and digestibility of noodles. The characteristics of highland barley flour, including proximate composition, pasting properties, and dough mixing behavior, were analyzed. The quality of the resulting noodles was evaluated through cooking and textural property analysis. The digestion characteristics of the noodles were determined to evaluate the starch hydrolysis rate and glycemic index (GI). Additionally, a correlation analysis was conducted among the proximate composition of highland barley flour, the characteristics of flour, and the quality of noodles. The results demonstrate that Chaiqing 1 exhibited superior performance in terms of flour quality and noodle texture compared to other varieties. The noodles produced from this variety possessed an outstanding texture, with moderate hardness and excellent elasticity. Additionally, its noodles also exhibited superior cooking resistance and low cooking loss. Nutritionally, the moderate estimated glycemic index (eGI) and high resistant starch (RS) content of Chaiqing 1 were beneficial for intestinal health. Ximalaya 22 showed good processing performance but slightly inferior texture, whereas Kunlun 14 had a high dietary fiber content, which resulted in noodles prone to breaking. Through a comprehensive variety comparison and screening, Chaiqing 1 emerged as the preferred choice for producing high-quality highland barley noodles. Furthermore, correlation analysis revealed that dietary fiber was significantly and positively correlated with water absorption, stability time (ST), and hardness (*p* < 0.01). Amylose content was associated with peak temperature and breakdown viscosity. This study provides valuable insights into the selection of highland barley varieties for noodle production.

## 1. Introduction

Highland barley (*Hordeum vulgare* L. var. *nudum Hook.* f.), belonging to thegramineous hordeum genus, is widely cultivated in high-altitude regions. It is rich in fiber, vitamins, and protein, with low sugar and fat content [1]. Furthermore, it is rich in various bioactive compounds such as flavonoids, polyphenols, and *β*-glucan [2]. These components contribute to its documented health benefits, such as anti-obesity [3], cholesterol reduction, blood sugar regulation, and immune modulation [4,5,6] effects, making it an attractive ingredient for the development of functional foods.

Noodles are among the most important staple foods in China and other Asian countries. Traditional noodles are primarily composed of wheat flour, water, and salt [7]. Relevant scientific studies have shown that after consuming wheat flour noodles (WFNs), fasting blood sugar levels, triglycerides, total cholesterol, and other indicators are significantly increased; thus, they are not suitable for patients with hyperglycemia and diabetes [8]. The primary advantage of highland barley in noodle formulations is its high *β*-glucan content, which effectively reduces the GI by slowing starch hydrolysis [3,9]. Additionally, highland barley contains a high level of dietary fiber compared to other functional grains, such as buckwheat and quinoa [10].

To address these limitations, recent research has focused on the development of composite noodles incorporating highland barley (*Hordeum vulgare* L.) and other nutrient-dense grains. For example, Ahmad et al. [11] demonstrated that blending fermented wheat starch with mung bean, sorghum, and millet flours enhanced dietary fiber, total phenolic content, and iron bioavailability, while simultaneously improving cooking stability and sensory attributes. Notably, Ding et al. [12] achieved a breakthrough by formulating noodles containing 93% highland barley flour, which exhibited a low glycemic index (GI ≤ 55) due to its slow carbohydrate digestibility. The impact of starch on GI is governed by its structural properties: amylose resists enzymatic hydrolysis due to its compact helical structure [13], whereas amylopectin is readily digested owing to accessible branching points [14]. Bioactive components like polyphenols [15] further attenuate starch digestion by increasing viscosity and inhibiting digestive enzymes [16].

Critically, current studies have predominantly focused on single highland barley varieties or simplified formulations [17,18], overlooking the systemic impact of inter-varietal compositional differences on product quality and metabolic outcomes. Genetic diversity [19] among highland barley varieties leads to significant compositional differences in total phenolic (62.26–609.66 mg/100 g), *β*-glucan (3.33–8.97%), and amylose (3.03–20.89%) [20]. These compositional disparities directly influence both the physicochemical properties of flour products and their physiological effects. For instance, *β*-glucan levels—a determinant of soluble fiber content—not only modulate dough rheology and textural attributes but also regulate postprandial lipid metabolism and glucose absorption rates [3]. Similarly, the ratio of amylose in grain starch influences noodle characteristics such as water absorption, optimum cooking time, and texture [21], a key predictor of GI [22]. Kaur et al. [23] confirmed that the difference in the composition of different wheat varieties of starch would significantly change the quality of bread. To address this gap, we systematically analyzed ten highland barley (*Hordeum vulgare* L.) varieties sourced from major production regions in China. These varieties were chosen for their representativeness and wide cultivation. This study focused on their approximate composition, pasting characteristics, and dough rheological properties. Subsequently, the quality attributes of noodles produced from these varieties were assessed through standardized cooking experiments and texture profile analysis (TPA) using a texture analyzer (Stable Micro Systems, Godalming, UK). Furthermore, an in vitro simulated digestion model was employed to quantify starch hydrolysis kinetics and calculate the eGI as indicators of digestibility. Multivariate correlation analysis revealed significant relationships (*p* < 0.05) between key compositional parameters, flour functional properties, and end-product quality metrics. These findings provide actionable insights for the variety-specific selection of highland barley in noodle manufacturing, enabling the optimization of processing parameters and the design of nutritionally tailored grain-based foods for populations with metabolic disorders.

## 2. Materials and Methods

### 2.1. Materials

The highland barley varieties used in this study were provided by the Qinghai Academy of Agriculture and Forestry Sciences (Table 1). Wheat flour with high gluten content was purchased from Xinliang Flour Co., Ltd., Xinxiang, China. Gluten flour was obtained from ZhongYu Food Co., Ltd., Binzhou China) All other chemicals and reagents used in this study were of analytical grade.

### 2.2. Preparation of Highland Barley Flour and Determination of Its Main Nutrients

The highland barley grains were screened, washed, dried, and subsequently ground with a cyclone mill (FOSS CT 293 Cyclotec™, Hillerød, Denmark). The resulting highland barley flour was sieved through an 80-mesh screen, and the yield was about 80% [24]. Finally, the highland barley flour was packed into polypropylene bags and stored at 4 °C. The moisture, ash, and protein contents of highland barley flour were determined according to GB 5009.3-2016 [25], GB 5009.4-2016 [26], and GB 5009.5-2016 [27]. The contents of total starch, amylose, total phenol, and total dietary fiber were determined using Megazyme assay kits (Megazyme Ireland, Bray Town, Ireland).

### 2.3. Preparation of Highland Barley Flour–Wheat Flour Composite Noodles

Highland barley flour cannot form dough independently in noodle preparation due to its absence of functional gluten proteins. Based on preliminary experiments, the optimal blend ratio was established as 50% highland barley flour, 42.5% wheat flour, and 7.5% gluten flour (*w*/*w*). All ingredients were mixed in a dough mixer (Guangzhou Nanxing Machinery Co., Ltd., Dongguan, China) for 3 min. After storage at 4 °C for 18 h. The dough was extruded using a noodle machine (Pasta extruder, Bühler Group, Switzerland) and sliced into noodles (length × width × thickness, 20 cm × 2 mm × 1 mm). The noodles were dried at 40 °C for 6 h.

### 2.4. Pasting Properties

A Rapid Viscosity Analyzer (RVA, Newport Science Corp., Melbourne, Australia) was employed to assess the pasting properties of composite flour, with modifications made according to Chen et al. [28]. Composite flour (3.00 ± 0.01 g) was mixed with water (25 ± 0.01 g) in an aluminum container. The testing procedure was as follows: the stirring speed was 160 rpm/min, the temperature was maintained at 50 °C for 1 min, then increased to 95 °C at a heating rate of 10 °C/min and held at this temperature for 2 min, and finally cooled to 50 °C at a rate of 10 °C/min and maintained at this temperature for 2 min.

### 2.5. Mixing Behavior

Mixolab (Chopin Technologies, Paris, France) was used to measure the mixing behavior of composite flour [29]. The torque (Nm), WAC (%), and ST (min) of dough at different forming stages were measured. Other moments of different periods (C1, C2, C3, C4, C5, Nm) were derived from the recorded curve. C5-C4 (Nm) and C1-C2 (Nm) were calculated. C3-C4 (Nm) and C5-C4 (Nm) were calculated.

### 2.6. Scanning Electron Microscopy and Color of Noodles

Scanning electron microscopy (SEM, Hitachi S-570, Japan High-tech International Trade Co., Ltd., Shanghai, China) was used to observe the microstructure of noodles. The dry noodles were cut into small pieces; soaked in a 2.5% glutaraldehyde solution for 2 h; then rinsed with phosphate buffer (0.1 mol/L, 4 °C) 4 times; eluted with 30%, 50%, 70%, 90%, and 100% ethanol step by step; then replaced with pure tert-butanol; and, finally, freeze-dried for 24 h [30]. The noodles were fixed on the platform with double-sided tape and sprayed with gold and the transverse microstructure was observed in the electron microscope chamber at 1500× magnification.

To accurately measure the color of the noodles, the noodles were ground and passed through a 40-mesh sieve before the experiment [31]. The *L** value, a* value, and b* value were measured using a color difference meter (Digieye 2.7, Verivide, Leicester, UK). For the a* value, red denoted a positive value and green denoted a negative value. For the b* value, yellow denoted a positive value and blue denoted a negative value [32].

### 2.7. Cooking Properties

Dry noodles (20 g) were weighed and placed in boiling water (500 mL). During cooking, the noodle strands were checked at 30 s intervals by squeezing the sample until the uncooked core of the noodle disappeared. This was considered the optimum cooking point, and the time recorded was the optimum cooking time [23]. After boiling the noodles to their optimum cooking time, they were transferred to room-temperature distilled water (25 °C) for 30 s. Subsequently, the complete length of the noodles was measured and the broken ratio was calculated. The gruel of the boiled noodle soup was collected for further analysis of cooking loss and cooking yield. The cooking yield was measured by Equation (1) [23]:(1)$$Cooking\;Yield\;(\%)=(m_{2}-m_{1})/m_{1}\times 100$$
where m_2_ represents the weight of the wet noodles after boiling and m_1_ represents the mass of the dry noodles before boiling.

### 2.8. Textural Properties

The textural properties of noodles were measured according to the method of Liu [33]. Noodles were boiled in water for the optimum cooking time and immediately transferred to cool water for 1 min. The water on the surface of noodles was wiped with paper and then the noodles were placed on the measuring table to measure textural properties using a texture analyzer (Stable Micro Systems, Godalming, UK). The TPA parameters were as follows: P/36R probe, test speed 1.0 mm/s, trigger force 20 g, and compression strain 70%. The shear parameters were as follows: A/LKB-F probe, test speed 0.8 mm/s, trigger force 3 g, and compression strain 50%. The parameters of the tensile assay were as follows: A/SPR probe, test speed 3 mm/s, trigger force 5 g, tensile distance 100 mm, and data acquisition frequency 500 p/s. The experiment was completed within 10 min and was repeated 6 times.

### 2.9. In Vitro Digestibility of Noodles

The assay was conducted according to the method in [34], with slight modifications. Noodles were mixed with 2 mL of distilled water and subjected to a boiling water bath for 20 min. After pasting, the mixture was cooled to 37 °C and then blended with 1 mL of *α*-amylase (50 U/mL) and three glass spheres for 2 min. Subsequently, 5 mL of pepsin solution (400 U/mL) was added to the mixture and agitated at 37 °C (300 rpm) for a duration of 30 min. Thereafter, 5 mL of 0.02 M NaOH, 20 mL of sodium acetate buffer (pH 6.0), and trypsin/amyloglucosidase were added to the mixture. Then, 1 mL of sample was taken out after 0 min, 10 min, 20 min, and up to 180 min of incubation. Next, the sample was combined with an equal volume of anhydrous ethanol before centrifugation at 2000× *g* for 5 min. The supernatant (0.1 mL) was subsequently mixed with 3 mL GOPOD in a water bath maintained at 50 °C for 20 min before measuring the absorbance at 510 nm. Rapidly digestible starch (RDS), slowly digestible starch (SDS), and RS contents were calculated using Equations (2)–(4). The starch digestion curve was modeled by applying a non-linear first-order rate equation, as described in Equation (5) [34]. The area under the curve (AUC) was calculated using Equation (6) [35]. The eGI was calculated according to Equation (7) [36]:(2)RDS (%)=(G20-FG) × 0.9/TS × 100(3)SDS (%)=(G120-G20) × 0.9/TS × 100(4)RS (%)=[TS-(RDS+SDS)] × 0.9/TS × 100(5)$$C_{t}=C_{\infty }\;(1-e^{-kt})$$(6)$$\mathrm{AUC}=C_{\infty }\;(t_{f}-t_{0})-(C_{\infty }/k)[(1-\exp-k(t_{f}-t_{0})]$$(7)eGI=37.91+0.549 × HI
where *C_∞_* is final/equilibrium concentration or digestibility, *k* reflects initial digestion rates, t (min) is the chosen time, t_f_ is the finaltime (180 min), t_0_ is the initial time (0 min), and hydrolysis index (HI) is the starchy foods.

### 2.10. Statistical Analysis

The assays were carried out at least three times and data were expressed as the mean ± standard deviation. SPSS 2024 (29.0.1.0) was used for variance analysis and Origin 2024 (v10.9.0.188) was used to construct graphs.

## 3. Results and Discussion

### 3.1. Proximate Composition of Highland Barley Flour

The proximate compositions of highland barley flour were comprehensively analyzed and are summarized in Table 2. The moisture content of the samples ranged from 7.68% to 9.62%, indicating relatively stable moisture levels. The average ash content of highland barley samples was 1.58%, ranging from 1.49% to 1.95%. The average protein content of highland barley samples was 11.01%, with a range of 9.06% to 13.07%. Among the highland barley varieties, Chaiqing 1 (13.70%), Kunlun 14 (13.13%), and Gankennuo 2 (12.82%) had protein contents higher than 12.00%, while Longzihei had the lowest protein content (9.07%). Although, highland barley flour could not form dough independently in noodle preparation due to its absence of functional gluten [37]. Sissons discovered that a high protein content not only enhanced the water absorption capacity (WAC) of dough but also exerted a substantial influence on the processing characteristics and quality attributes of noodles [38]. As the principal constituent of highland barley, total starch exhibited an average content of 55.34%, with the variation ranging from 51.09% to 61.27%. Zangqing 3000 demonstrated the highest total starch content (61.27%), potentially affecting starch pasting properties during noodle preparation. In this study, the average amylose content was 23.08%, the distribution range was 7.63–31.66%, and the proportion of amylose in total starch was 13.25% to 52.48%. Total dietary fiber is an important nutrient in whole-grain foods. The average content of total dietary fiber was 19.81%, with a range of 15.14% to 27.78%. Kunlun 14 had the highest total dietary fiber (27.78%), significantly higher than Beiqing 8 (15.14%). The average content of *β*-glucan was 4.80%, with a range of 4.24% to 5.59%. As shown in Table 2, the total phenol and total dietary fiber contents in wheat flour were significantly lower than those in highland barley flour (*p* < 0.05). The average total phenol content of highland barley samples was 2.52 mg/g, with a variation range of 2.05 mg/g to 3.36 mg/g. The obtained results were largely in agreement with the measured contents of the relevant components in barley reported by Zhu et al. [39].

### 3.2. Pasting Properties of Composite Flour

The RVA assay reflected the changes in the viscosity of grain flour or starch solutions during heating and cooling and predicted the quality of the noodles [40,41,42,43]. During the pasting of flour, water entered the crystal structure of starch and broke the hydrogen bonds; this process caused starch to change from an ordered to a disordered structure. As shown in Table 3, the peak viscosity, which indicated the maximum swelling capacity of the starch granules during heating, was highest for Gankennuo 2 (1963.50 mPa·s), likely due to its high amylopectin content, as amylopectin promotes rapid water absorption and granule swelling [44]. However, Gankennuo 2 also exhibited the highest breakdown viscosity (933.50 mPa·s), indicating low thermal stability due to its high amylopectin content. Extensive branching weakens starch crystalline regions, promoting granule disintegration under shear stress [45]. In contrast, Kunlun 14 demonstrated the lowest breakdown viscosity (471.50 mPa·s) and peak viscosity (1130.00 mPa·s), suggesting the best thermal stability, which is advantageous for maintaining noodle integrity during cooking. The trough viscosity reflected the shear resistance of the starch paste at high temperatures. The trough viscosity value of Gankennuo 2 (1013.00 mPa·s) was significantly higher than that of other varieties, which might have been caused by its higher amylopectin content. The final viscosity, which reflected the ability of the starch to form a viscous paste after cooling [46], was higher for Ximalaya 22 (1833.50 mPa·s) and Zangqing 3000 (1813.00 mPa·s). This can be attributed to their high total starch content, which enhanced water absorption and granule swelling [47]. The reduced final viscosity observed in Kunlun 14 compared to Gankennuo 2 was because in addition to the influence of amylose, the final viscosity was also affected by the total starch content and other components [22], as well as the distribution of amylopectin chain length [48]. Setback viscosity, associated with starch retrogradation, was highest in Ximalaya 22 (880.50 mPa·s). High setback viscosity values correlate with rapid starch recrystallization during cooling, which is primarily driven by amylose realignment [49]. Compared with other varieties, Ximalaya 22 had a relatively high amylose content (31.66%, as shown in Table 2). A study by Ribeiro [50] indicated that a higher amylose content could promote more extensive amylose realignment during the cooling process. This leads to the more rapid formation of a rigid starch-based network structure, resulting in a higher setback viscosity. In contrast, Zangqing 2000 had the lowest setback viscosity (702.50 mPa·s) and final viscosity (1294.00 mPa·s), indicating a reduced retrogradation tendency [51]. These properties aligned with Zangqing 2000, which had lower cooking loss (5.58%), as reduced retrogradation minimized starch leaching during cooking [52]. The peak time, which reflected the resistance of starch granules to swelling, was longest for Beiqing 8 and Ximalaya 22 (6.20 min), indicating greater thermal resistance. The peak temperature, which indicated the temperature at which maximum viscosity was achieved, was highest for Beiqing 8 (90.43 °C), further supporting its thermal stability. However, the lowest peak temperature of Gankennuo 2 was 73.50 °C, which may be attributed to its significantly low amylose content (7.63%, as shown in Table 2). A reduction in amylose content causes the crystalline structure of starch particles to become unstable, making it easier to gelatinize at lower temperatures. Although the amylose content of Longzihei and Beiqing 8 was similar, there was a significant difference in their peak temperatures. This was because the peak temperature was not only affected by the content of amylose but also by the total starch content, other components such as proteins and lipids [22], and the distribution of amylopectin chain length [48]. The relationship between amylose content and peak temperature did not appear to hold for Longzihei and Ximalaya 22. This was because the peak temperature of starch in the samples was influenced not only by amylose content but also by other critical factors, such as the molecular structure of starch granules, crystallinity characteristics, and interactions with non-starch components [22].

### 3.3. The Mixing Behavior of Composite Flour

The Mixolab test can provide information on dough mixing behavior. In the test, the torque generated by the dough passing through two kneading arms was recorded to study the mixing and pasting properties of the composite flour. WAC, (%), ST, (min), and C2 parameters were used to predict the gluten quality. The C3, C5, amylase activity (C3-C4, Nm), and starch aging (C5-C4, Nm) parameters indicated the starch pasting performance of the composite flour system during heating and cooling [53]. The WAC is a critical parameter reflecting the ability of composite flour to absorb water during dough formation. As shown in Table 4, the WAC values ranged from 71.30% to 77.07%, with Kunlun 14 exhibiting the highest WAC (77.07%). This high WAC can be attributed to its high total dietary fiber content (27.78%, Table 2), which contained hydrophilic groups that enhanced water retention by forming hydrogen bonds with water molecules [54]. Nevertheless, when the dietary fiber content was too high, it competed with starch and gluten for water absorption. In the case of Kunlun 14, although its high dietary fiber content contributed to high WAC, it also had the potential to disrupt gluten network formation to some extent [55,56,57]. ST is an indicator of dough strength, referred to as the “tolerance” of flour to over-mixing and under-mixing. Weak gluten flours generally have a shorter stabilization time than strong gluten flours [55]. A longer ST suggests better dough stability and stronger gluten networks. Chaiqing 1 had the longest ST (9.92 min), which can be attributed to its balanced protein (13.70%) and *β*-glucan (5.59%) contents. These components synergistically strengthen the dough matrix, with protein acting as a structural scaffold and *β*-glucan forming a viscous network that stabilizes starch granules during mixing [51]. However, different highland barley varieties showed significant differences in dough stability. Beiqing 8 exhibited the shortest ST (8.28 min), whereas Chaiqing 1 demonstrated the longest ST (9.92 min). C2 represented the loss of dough consistency when exposed to physical, mechanical, and thermal stress [58]. A lower C2 value indicated stronger processing ability and better dough stability. Ximalaya 22 had the lowest C2 value (0.42 Nm), suggesting its superior processing characteristics. During the heating process of the Mixolab test, C3 indicated the degree of starch pasting, C3-C4 represented amylose activity [33], and C5 and C5-C4 mainly reflected the properties of wheat starch in dough during thermal treatment [59]. The C3 value of Gankennuo 2 was the lowest, indicating its lowest stability and heat resistance. Its pasting characteristics (peak temperature = 73.50 °C, peak time = 5.84 min) were consistent with the RVA results. This suggests that Gankennuo 2 was more prone to starch degradation during processing, which may have affected the final product quality. On the other hand, Ximalaya 22 exhibited significantly lower C5 and C5-C4 values, indicating that its composite flour was less susceptible to aging and more conducive to extending the shelf life of flour products [58]. This is crucial for developing noodles with better storage stability. The higher the C5-C4 value, the higher the degree of starch aging. The C5-C4 of Zangqing 3000 was the highest, indicating that the pasting degree of starch was the highest, and the product was more easily aged, which was consistent with the results of the RVA. The C3-C4 parameter (amylase activity) and C5-C4 (anti-aging effect) further characterized starch behavior. The C3-C4 value of Beiqing 8 was significantly lower than that of other varieties, at only 0.05 Nm. This suggests that its amylase activity and the degree of starch degradation were the lowest during the heating process. This finding is consistent with the RVA peak time (6.20 min, Table 3), indicating that the starch granules in dough of this variety can maintain high integrity and exhibit superior thermal stability during heating.

### 3.4. SEM Observation and Color of Noodles

The microstructure of noodles, as shown in Figure 1, revealed critical differences in gluten network integrity and starch–protein interactions among the highland barley varieties when observed via SEM analysis. In WFN (Figure 1K), smooth starch granules were uniformly embedded within a continuous gluten matrix, forming a compact network. In contrast, composite flour noodles exhibited discontinuous structures due to the interference of total dietary fiber. For instance, Zangqing 3000 (Figure 1J) displayed excessive starch aggregation, likely owing to its high total starch content (61.24%, Table 2). Such compactness may explain its high peak viscosity (1625 mPa·s, Table 3) but poor thermal stability (breakdown viscosity = 670.50 mPa·s). The densely packed starch granules in Zangqing 3000 noodles were prone to rapid swelling and rupture during heating, which led to a higher cooking loss during cooking. The densely packed starch granules in Zangqing 3000 noodles tended to swell and rupture rapidly during heating, leading to higher cooking loss. This phenomenon is consistent with the findings of Wang et al. [60], who confirmed that the structural collapse of starch granules—caused by the melting of the crystalline zone, the dissociation of double helices, and the breakage of hydrogen bonds—is a critical step in gelatinization. Starch systems with closely arranged granules and a high proportion of amorphous regions are more susceptible to rapid swelling and cracking during heating, thereby significantly increasing cooking loss. On the other hand, Chaiqing 1 (Figure 1A) demonstrated a balanced structure with moderate porosity and fewer cracks, consistent with its superior cooking yield (136.23%, Table 5). Notably, Kunlun 14 (Figure 1D) exhibited large cracks (indicated by yellow circles) and irregular pores, consistent with its high broken ratio (11.67%, Table 5). This structural fragility may have resulted from its high total dietary fiber content (27.78%, Table 2), which competed with gluten for water absorption, leading to incomplete network formation [57].

Noodles with a brighter color are generally more acceptable to consumers. Wheat flour contains yellow pigments such as lutein and carotene, so WFNs appeared pale yellow [42]. As illustrated in Table 6, compared with WFNs, the composite noodles exhibited lower *L** values (indicating brightness) and higher a* (red–green) and b* (yellow–blue) values. This color shift can be attributed to the presence of phenolic compounds and insoluble total dietary fiber in highland barley flour. These components may undergo oxidation or interact with starch–protein complexes during processing, leading to darker hues [61]. Among these composite noodles, the *L** value (85.62) of Chaiqing 1 was the highest, while the a* and b* values were relatively lower, indicating that the color of Chaiqing 1 composite noodles was closer to that of the WFNs. This suggests that Chaiqing 1 may be more acceptable to consumers who prefer the traditional appearance of WFNs.

### 3.5. Cooking Properties of Noodles

To better understand the effects of different varieties of highland barley on the quality of the final product, the cooking properties of the noodles prepared with different varieties of composite flour were measured. The optimal cooking time varied due to differences in the composition of the highland barley varieties (*p* < 0.05), ranging from 3.46 min to 5.07 min. Specifically, the optimal cooking time for Zangqing 3000 was the longest at 5.07 min. In comparison, the optimal cooking time for Kunlun 14 was 3.46 min, which was 31.8% shorter than that of Zangqing 3000. Most varieties with C2 < 0.50 Nm (e.g., Chaiqing 1, Gankennuo 2, Ximalaya 22, Zangqing 2000, and Zangqing 3000) showed a 0% broken ratio, indicating better dough stability and resistance to mechanical stress during cooking. However, Zangqing 25 (C2 = 0.53 Nm, broken ratio = 5.00%) exhibited higher fragility. Kunlun 14 exhibited the highest broken ratio (11.67%), likely due to its high total dietary fiber content (27.78%, Table 2) interfering with gluten network formation, leading to structural fragility. The cooking loss reflected the structural integrity of the noodles during cooking [62] and was related to the release or dissolution of solids from noodles during cooking, which is undesirable in noodle making [63]. Composite noodles exhibited higher cooking loss (5.55–6.64%) than WFNs (5.34%); Kunlun 14 showed the highest cooking loss (6.64%), likely due to its dietary fiber (27.78%, Table 2) competing with starch for water absorption, weakening the gluten network and promoting starch leaching during cooking [46]. The cooking yield of composite noodles ranged from 119.97% to 136.23%. In contrast, Chaiqing 1 exhibited the lowest cooking loss (5.55%), which can be attributed to its balanced (5.59%) and protein (13.70%) contents, forming a cohesive network that enhanced dough elasticity and resistance to shear stress, as evidenced by its moderate porosity observed in SEM images.

### 3.6. Textural Properties of Noodles

Textural parameters can often be considered essential indicators that affect taste and noodle acceptance. The textural properties (TPA, shear, and tensile parameters) of the composite noodles and WFNs are presented in Table 7. Hardness is an important indicator of the quality of noodles, but an excessively high hardness is accompanied by an increase in brittleness. Composite noodles generally exhibited higher hardness and tensile force than WFNs. This was primarily attributed to the higher dietary fiber content in highland barley, which restricted the network formation of gluten proteins in the dough. Consequently, this led to a denser noodle structure and increased hardness [33]. Ling [64] discovered that a higher dietary fiber content inhibited the network formation of gluten proteins in dough, leading to a denser noodle texture and greater hardness. The hardness of Kunlun 14 was the highest while that of Gankennuo 2 was the lowest. Adhesiveness, which reflected the work required to separate the noodles from the surface of the tongue or teeth, was significantly lower in composite noodles than in WFNs (*p* < 0.05). This can be explained by the fact that wheat flour forms a denser gluten network, which is more adhesive than the gluten-free network formed by highland barley [65]. Among the composite noodles, Chaiqing 1 exhibited the lowest adhesiveness, which may have been related to its low TS content because starch can increase the adhesive properties of noodles [66]. The springiness and chewiness of the noodles were similar between the WFNs and composite noodles. Chaiqing 1, Ximalaya 22, Kunlun 15, Zangqing 2000, and Zangqing 25 showed higher springiness (>98 g·s). For Chaiqing 1, this may have resulted from its balanced protein and *β*-glucan contents forming elastic networks [67]. Notably, the rest achieved this despite potential differences in protein composition, suggesting that additional factors (e.g., starch retrogradation) contribute to elasticity.

Tensile force measured the resistance of the noodles to breaking under tension. The shear force and tensile force of highland barley noodles were significantly higher than those of WFNs, indicating superior mechanical properties. In these composite noodles, Kunlun 14 exhibited the highest shearing force and tensile force, which can be attributed to the total dietary fiber interacting with proteins and starch, enhancing the shear resistance of the noodles [68,69] and thus affecting the tensile properties [55]. Generally, noodles with low adhesiveness [70] and high springiness, chewiness, and tensile strength exhibited a smooth taste [71]. Among the composite noodles, Chaiqing 1 presented the highest springiness, relatively high chewiness and tensile force, and lowest adhesiveness value, so the texture quality of Chaiqing 1 was the best.

### 3.7. Digestive Properties of Noodles

To further evaluate a more comprehensive analysis of the nutritional function of noodles, the starch digestibility of noodles was analyzed. GI is a physiological metric that quantifies the incremental area under the blood glucose curve following the ingestion of a test food relative to the area under the curve for an equivalent amount of carbohydrate from a reference food (white bread or glucose) consumed by the same individual [72]. Foods are classified based on their GI values: those with a GI greater than 70 are categorized as high-GI foods, those with a GI between 55 and 70 are considered medium-GI foods, and those with a GI less than 55 are designated as low-GI foods [72]. The starch hydrolysis rate is a critical parameter that indicates how quickly starch is broken down into simpler sugars during digestion. As shown in Figure 2A, the digestion profile of WFNs exhibited a steadily increasing trend, and a similar trend was observed in composite noodles. However, the starch hydrolysis rate of composite noodles was significantly lower than that of WFNs. This was mainly due to the high total dietary fiber content in highland barley flour. Dietary fiber can physically entangle with starch molecules, creating a barrier that hinders the access of *α*-amylase to the starch substrate [73]. In addition, the presence of certain bioactive compounds in highland barley, such as polyphenols, may also have an inhibitory effect on *α*-amylase activity, further contributing to the reduced starch hydrolysis rate [5]. Typically, consuming excessive amounts of RDS can lead to significant postprandial blood glucose spikes, negatively impacting blood glucose stability. In contrast, consuming SDS supports the maintenance of postprandial blood glucose balance, and RS intake can aid in preventing colon cancer and enhancing gut health [74]. As shown in Figure 2B, the RDS and SDS contents of composite noodles were lower than those of WFNs, whereas the RS content was significantly (*p* < 0.05) higher than that of WFNs. RS exhibited significantly higher levels in Kunlun 14, which might be why Kunlun 14 exhibited large cracks and why its stomata were irregular (Figure 1D), which was unconducive to the entry of enzymes into the sample [19]. In addition, the dietary fiber content of Kunlun 14 was 27.98%. Dietary fiber can form a physical barrier, hindering the hydrolysis of starch by digestive enzymes, while increasing the viscosity of intestinal contents and delaying the digestion process of starch [73]. Conversely, the RS content of WFNs was the lowest and the eGI was the highest, which suggests that the RS content was highly significantly negatively correlated with eGI (*p* < 0.01, Figure 3). High dietary fiber (e.g., Kunlun 14 = 27.78%) increased RS formation by physically entrapping starch granules, thereby restricting the access of digestive enzymes [73]; this finding was consistent with the significant positive correlation observed between RS content and total dietary fiber (Figure 3). Low amylose content (e.g., Gankennuo 2 = 7.63%) accelerated starch digestion by exposing amylopectin branching points, facilitating rapid enzymatic hydrolysis and elevating RDS levels [75]. This result was consistent with the content of RDS being significantly negatively correlated with amylose content (*p* < 0.01), as shown in Figure 3.

In order to further analyze the digestive characteristics of composite noodles, the first-order non-linear fitting equation model was used to calculate the kinetic parameters of starch digestion. As shown in Table 8, the eGI values of highland barley composite noodles were significantly lower than those of WFNs (*p* < 0.05). Among the tested varieties, Kunlun 14 exhibited the lowest eGI value (65.04), whereas Gankennuo 2 exhibited the highest value (78.24). This variation may be attributed to differences in the dietary fiber content. As reported by Zhang et al. [76], eGI was significantly negatively correlated with total dietary fiber content (Figure 3). Dietary fiber can also interact with water, reducing the effective water availability for starch gelatinization during noodle processing. This makes the starch less accessible to enzymatic hydrolysis, thereby slowing down the starch hydrolysis rate. The composite noodles prepared from five varieties (Chaiqing 1, Kunlun 14, Longzihei, Ximalaya 22, and Zangqing 2000) qualified as medium-GI foods (GI 55-70). Notably, Beiqing 8, Gankennuo 2, Kunlun 15, Zangqing 25, and Zangqing 3000 approached high-GI classification. The parameter *C_∞_* was derived from fitting in vitro digestion data to a first-order kinetic model. Studies have shown that eGI is influenced by both *C_∞_* and the kinetic constant *k* [35,36]. Kunlun 14 had the lowest eGI (65.04), indicating that fibers might adsorb starch and reduce enzymatic hydrolysis [77]. These results demonstrate that high-fiber varieties (e.g., Kunlun 14 and Chaiqing 1) are optimal for slow-digesting noodles.

### 3.8. Pearson’s Correlation Analysis

Pearson’s correlation analysis revealed a significant correlation between the proximate composition of highland barley, flour characteristics, and noodle quality. As depicted in Figure 3, the mixing behavior of the dough was primarily influenced by total dietary fiber and amylose. A significant positive correlation was observed between total dietary fiber and WAC as well as ST (*p* < 0.01). The hydrophilic groups of dietary fiber enhance the water-holding capacity of dough. However, excessive dietary fiber may compete with gluten proteins for water, thereby disrupting the continuity of the gluten network and leading to an increased broken ratio of noodles, as observed in Kunlun 14 (Table 5). Amylose was positively correlated with C3 (*p* < 0.001), which may have been because the tight helical structure of amylose requires a higher temperature to disrupt the crystal structure [22]. For instance, Gankennuo 2, which had a low amylose content (7.63%), exhibited the lowest peak temperature (73.50 °C), confirming this mechanism. *β*-glucan was positively correlated with C5-C4 (*p* < 0.05), indicating that it delayed starch retrogradation by forming a viscous network [3]; this was consistent with the lower cooking loss (5.55%) and better elasticity (98.84 g·s) observed for Chaiqing 1 noodles. The mixing behavior of dough is closely associated with gluten content, which is an essential index for evaluating dough formation. Pasting characteristics reflect the process of starch swelling at high temperatures to form a gel, which is closely linked to the cooking quality of the flour-based products. The pasting characteristics of highland barley flour were mainly affected by amylose, shown in Figure 3. Amylose exhibited a negative correlation with breakdown viscosity (*p* < 0.01) as it inhibited the excessive expansion and rupture of starch granules [22]. Consistent with this, Table 2 shows that Gankennuo2, which had a high content of amylopectin, exhibited the highest breakdown viscosity (933.50 mPa·s). The peak temperature was negatively correlated with the total phenol content (*p* < 0.01), likely due to the ability of polyphenols to stabilize the starch crystal structure through hydrogen bonding, thereby delaying the gelatinization initiation temperature [78].

In terms of noodle quality, ash content was negatively correlated with shear force. This was likely due to the interaction between minerals in the ash and proteins in the dough, which formed complexes and interfered with protein network formation, thereby reducing dough elasticity and strength and consequently decreasing shear force. Adhesiveness was negatively correlated with the total phenol content and positively correlated with the amylose content (*p* < 0.01). Cooking yield was positively correlated with *β*-glucan content (*p* < 0.01), suggesting that higher *β*-glucan levels may enhance the structural integrity and water retention capacity of noodles during cooking [14]. The total dietary fiber content exhibited a significant positive correlation with hardness (*p* < 0.01), likely due to its physical reinforcing effect, which led to a more compact dough structure. For example, Kunlun 14, having the highest fiber content (27.78%), demonstrated the greatest hardness at 8618.90 g. Total starch content exhibited a significant positive correlation with optimal cooking time (*p* < 0.05), suggesting that higher starch content typically necessitates longer cooking times for complete gelatinization. For instance, Zangqing 3000, with a high starch content of 61.27%, required a longer optimal cooking time of 5.07 min (Table 5). The content of RDS was significantly negatively correlated with amylose content (*p* < 0.01) [79]. The RS content was significantly positively correlated with total dietary fiber (*p* < 0.01), a finding also confirmed by Fatimah [80] and Miller et al. [81]. The eGI was significantly negatively correlated with total dietary fiber content (*p* < 0.01), which is consistent with the results reported by Bakar [82]. The content of RS was significantly negatively correlated with eGI (*p* < 0.01) [83]. The factors influencing noodle quality are highly complex and are related not only to the proximate composition of flour but to the mixing behavior of the dough and pasting characteristics, with evaluation criteria often varying.

## 4. Conclusions

The physicochemical characteristics of highland barley varieties significantly affected the quality, digestion characteristics, and processing adaptability of the noodles. Specifically, the protein content ranged from 9.07% to 13.70%, total starch content from 51.09% to 61.27%, amylose content from 7.63% to 31.66%, total dietary fiber content from 15.14% to 27.78%, *β*-glucan content from 4.24% to 5.59%, and total phenol content from 2.05 to 3.36 mg/g. These variations in proximate components led to differences in the physicochemical properties of each composite flour. The mixing behavior and pasting properties of barley composite flour were primarily influenced by dietary fiber and amylose. Dietary fiber was significantly positively correlated with stability time and water absorption capacity (*p* < 0.01), whereas amylose content was significantly positively correlated with peak temperature and C3 (*p* < 0.01). The optimal cooking time of noodles was closely related to the total starch content of the highland barley. Ash content was negatively correlated with shear force (*p* < 0.05), adhesiveness was significantly positively correlated with amylose content (*p* < 0.01), hardness was significantly positively correlated with total dietary fiber (*p* < 0.01), and *β*-glucan was positively correlated with cooking yield and C5-C4 (*p* < 0.05). SEM revealed that Chaiqing 1 had a relatively uniform and complete network structure, whereas Zangqing 3000 exhibited excessive tightness with agglomeration and caking, resulting in an unclear network structure. Kunlun 14 displayed evident cracks due to the interference of dietary fiber. Highland barley noodles generally had a darker color, with Chaiqing 1 being the closest in color to WFN. In terms of the cooking properties of the noodles, Chaiqing 1 and Zangqing 3000 demonstrated the best quality, followed by Ximalaya 22. Regarding texture characteristics, Chaiqing 1 ranked first, followed by Ximalaya 22. The digestive characteristics of the noodles made from different highland barley varieties varied significantly. Noodles prepared from Chaiqing 1, Kunlun 14, Longzihei, Ximalaya 22, and Zangqing 2000 were classified as medium-GI foods, with Kunlun 14 showing the lowest eGI value.

In conclusion, compared with other tested highland barley varieties, Chaiqing 1 possessed the best quality based on the three aspects: (1) product performance (better cooking and texture qualities), (2) processing adaptability (longer stabilization time and an ideal C2 value), and (3) nutritional quality (high protein and *β*-glucan contents). As a result, it is regarded as the perfect raw ingredient for making premium barley noodles. Ximalaya 22 ranked second, while the processability of the other varieties was relatively inferior.

## Figures and Tables

**Figure 1 foods-14-02163-f001:**
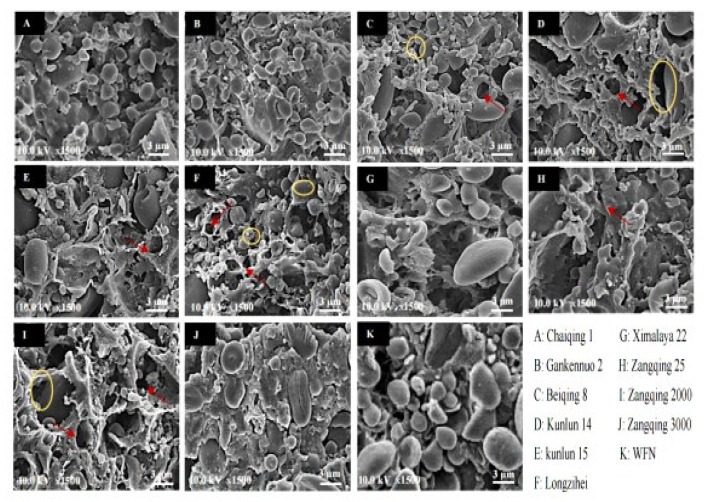
Scanning electron micrographs of noodles prepared from wheat flour and different highland barley varieties at magnifications of 1500×. Note: the red arrow marks irregular surface pores; the yellow circle denotes cracks.

**Figure 2 foods-14-02163-f002:**
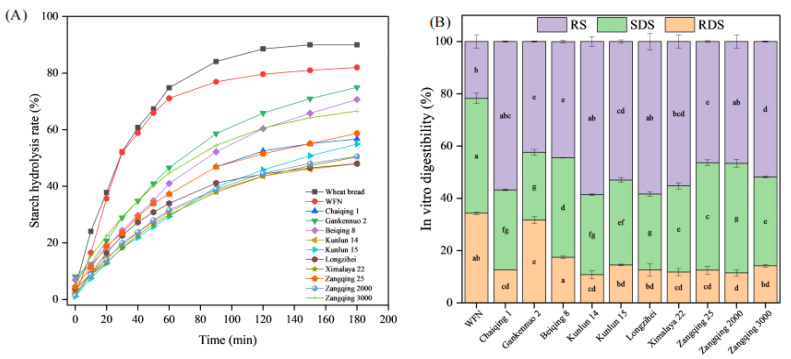
Study on in vitro digestibility of noodles prepared from wheat flour and different highland barley varieties (**A**). RDS, SDS, and RS content of noodles prepared from wheat flour and different highland barley varieties (**B**). The contents of SDS, RDS and RS among different highland barley varieties are marked with different small letters (a, b, c, d, e, f, g, h) for the expression of the significant difference (*p* < 0.05).

**Figure 3 foods-14-02163-f003:**
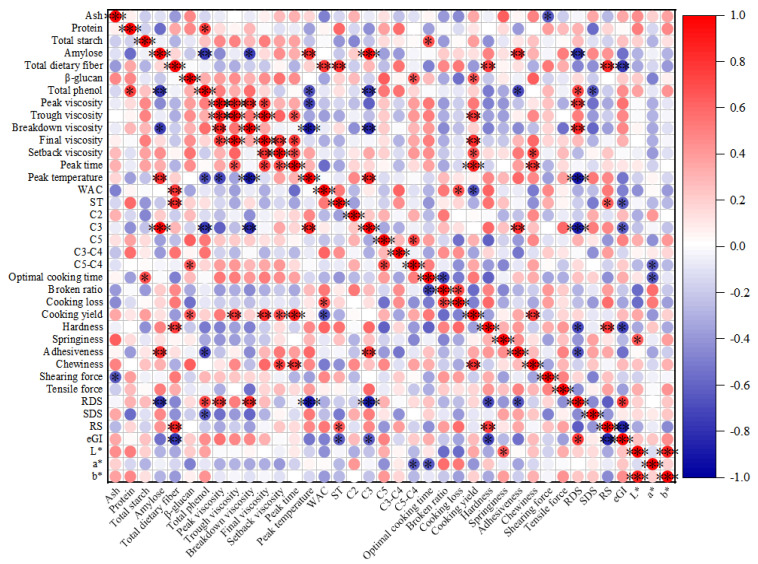
Pearson’s correlation between proximate composition of highland barley and flour characteristics and noodle quality. Note: *, *p* ≤ 0.05; **, *p* ≤ 0.01; ***, *p* ≤ 0.001.

**Table 1 foods-14-02163-t001:** Information about ten highland barley varieties used in this study.

Variety	Planting Area	Harvested Year	Variety	Planting Area	Harvested Year
Chaiqing 1	Qinghai	2022	Longzihei	Tibet	2022
Gankennuo 2	Gansu	2022	Ximalaya 22	Tibet	2022
Beiqing 8	Qinghai	2022	Zangqing 25	Tibet	2022
Kunlun 14	Qinghai	2022	Zangqing 2000	Tibet	2022
Kunlun 15	Qinghai	2022	Zangqing 3000	Tibet	2022

**Table 2 foods-14-02163-t002:** Proximate compositions of highland barley flour prepared from different highland barley varieties.

Sample	Moisture (%)	Ash (%)	Protein (%)	Total Starch (%)	Amylose (%)	Total Dietary Fiber (%)	*β*-Glucan (%)	Total Phenol (mg/g)
Chaiqing 1	8.25 ± 0.21 ^bc^	1.83 ± 0.01 ^abc^	13.70 ± 0.27 ^a^	51.09 ± 0.54 ^fg^	24.42 ± 0.01 ^e^	24.87 ± 0.64 ^b^	5.59 ± 0.02 ^a^	2.59 ± 0.07 ^de^
Gankennuo 2	9.58 ± 1.17 ^a^	1.62 ± 0.01 ^cd^	12.82 ± 0.08 ^b^	57.58 ± 0.02 ^c^	7.63 ± 0.14 ^h^	18.68 ± 0.52 ^e^	4.82 ± 0.14 ^bc^	3.36 ± 0.01 ^a^
Beiqing 8	8.51 ± 0.40 ^abc^	1.71 ± 0.00 ^bcd^	9.76 ± 0.05 ^d^	54.53 ± 1.40 ^def^	27.00 ± 0.26 ^c^	15.14 ± 0.21 ^f^	4.87 ± 0.14 ^bc^	2.58 ± 0.21 ^de^
Kunlun 14	7.68 ± 0.39 ^c^	1.49 ± 0.01 ^d^	13.13 ± 0.27 ^a^	53.54 ± 1.29 ^ef^	22.46 ± 0.48 ^f^	27.78 ± 0.01 ^a^	4.40 ± 0.24 ^de^	2.86 ± 0.02 ^bc^
Kunlun 15	8.50 ± 0.95 ^abc^	1.95 ± 0.04 ^a^	11.10 ± 0.18 ^c^	54.58 ± 1.41 ^def^	17.52 ± 0.11 ^g^	20.22 ± 0.98 ^d^	4.90 ± 0.15 ^b^	2.55 ± 0.36 ^de^
Longzihei	9.62 ± 1.19 ^a^	1.49 ± 0.13 ^d^	9.07 ± 0.08 ^d^	53.12 ± 0.57 ^e^	27.19 ± 0.26 ^c^	25.98 ± 0.80 ^b^	4.67 ± 0.08 ^cd^	2.14 ± 0.04 ^fg^
Ximalaya 22	8.09 ± 0.28 ^c^	1.61 ± 0.01 ^cd^	9.50 ± 0.12 ^d^	60.33 ± 0.04 ^b^	31.66 ± 0.71 ^a^	20.99 ± 0.20 ^cd^	4.24 ± 0.06 ^de^	2.19 ± 0.04 ^fg^
Zangqing 25	9.45 ± 0.25 ^ab^	1.90 ± 0.16 ^ab^	9.87 ± 0.03 ^d^	54.36 ± 0.28 ^def^	25.51 ± 0.25 ^d^	20.32 ± 0.07 ^d^	4.80 ± 0.14 ^bc^	2.08 ± 0.06 ^g^
Zangqing 2000	9.51 ± 0.27 ^ab^	1.64 ± 0.13 ^cd^	9.50 ± 0.08 ^d^	53.03 ± 0.03 ^f^	24.32 ± 0.01 ^e^	20.43 ± 0.20 ^d^	4.37 ± 0.06 ^de^	2.05 ± 0.24 ^g^
Zangqing 3000	8.70 ± 0.38 ^abc^	1.64 ± 0.01 ^cd^	11.02 ± 0.18 ^c^	61.27 ± 0.88 ^b^	23.08 ± 0.22 ^f^	21.88 ± 0.45 ^c^	5.31 ± 0.08 ^a^	2.77 ± 0.01 ^bc^
Mean value	8.79	1.58	10.95	55.34	23.08	19.81	4.80	2.52
Range	7.68~9.62	1.49~1.95	9.07~13.70	51.09~61.27	7.63~31.66	15.14~27.78	4.24~5.59	2.05~3.36
Wheat flour	9.49 ± 0.19 ^ab^	0.47 ± 0.04 ^e^	9.73 ± 0.17 ^d^	77.35 ± 0.37 ^a^	29.18 ± 0.24 ^b^	1.67 ± 0.03 ^g^	0.15 ± 0.04 ^f^	1.89 ± 0.34 ^h^

Note: % indicates the mass ratio; components other than moisture were measured on a dry basis. Results are presented as means ± standard deviations (*n* = 3). Values in the same column with different letters were different significantly (*p* < 0.05).

**Table 3 foods-14-02163-t003:** Pasting properties of composite flour from different highland barley varieties.

Sample	Peak Viscosity (mPa·s)	Trough Viscosity (mPa·s)	Breakdown Viscosity (mPa·s)	Final Viscosity (mPa·s)	Setback Viscosity (mPa·s)	Peak Time (min)	Peak Temperature (°C)
Chaiqing 1	1510.00 ± 46.67 ^d^	926.00 ± 15.56 ^c^	638.00 ± 45.25 ^bc^	1762.00 ± 22.63 ^b^	839.50 ± 2.12 ^bc^	6.13 ± 0.00 ^b^	88.78 ± 0.04 ^c^
Gankennuo 2	1963.50 ± 27.58 ^a^	1013.00 ± 4.24 ^a^	933.50 ± 7.78 ^a^	1731.00 ± 18.38 ^c^	732.00 ± 5.66 ^g^	5.84 ± 0.05 ^e^	73.50 ± 0.00 ^e^
Beiqing 8	1473.50 ± 10.61 ^d^	947.50 ± 9.19 ^bc^	526.00 ± 1.41 ^e^	1730.00 ± 12.72 ^c^	782.50 ± 3.54 ^e^	6.20 ± 0.00 ^a^	90.43 ± 0.04 ^a^
Kunlun 14	1130.00 ± 7.07 ^g^	658.50 ± 2.12 ^g^	471.50 ± 9.12 ^f^	1401.00 ± 9.90 ^f^	729.00 ± 7.07 ^g^	5.73 ± 0.00 ^f^	88.85 ± 0.00 ^b^
Kunlun 15	1326.50 ± 19.09 ^f^	734.50 ± 2.12 ^f^	614.50 ± 14.85 ^c^	1496.50 ± 2.12 ^e^	762.00 ± 4.24 ^f^	6.07 ± 0.00 ^c^	88.85 ± 0.00 ^b^
Longzihei	1559.00 ± 2.83 ^c^	888.00 ± 12.73 ^d^	665.00 ± 1.41 ^b^	1671.50 ± 10.61 ^d^	799.00 ± 1.41 ^d^	5.93 ± 0.00 ^d^	88.05 ± 0.00 ^d^
Ximalaya 22	1510.50 ± 10.61 ^d^	947.00 ± 5.56 ^bc^	566.00 ± 0.71 ^d^	1833.50 ± 4.95 ^a^	828.00 ± 1.41 ^c^	6.20 ± 0.00 ^a^	88.78 ± 0.04 ^c^
Zangqing 25	1413.50 ± 9.19 ^e^	853.50 ± 3.54 ^e^	556.00 ± 0.00 ^de^	1688.50 ± 2.12 ^d^	845.00 ± 8.49 ^b^	5.87 ± 0.00 ^e^	88.78 ± 0.04 ^c^
Zangqing 2000	1148.00 ± 1.41 ^g^	593.50 ± 13.43 ^h^	569.00 ± 5.66 ^d^	1294.00 ± 8.49 ^g^	702.50 ± 2.12 ^h^	5.64 ± 0.05 ^g^	88.05 ± 0.00 ^d^
Zangqing 3000	1625.00 ± 11.31 ^b^	964.00 ± 1.41 ^b^	670.50 ± 0.71 ^b^	1813.00 ± 7.07 ^a^	880.50 ± 7.78 ^a^	6.13 ± 0.00 ^b^	88.80 ± 0.00 ^bc^

Results are presented as means ± standard deviations (*n* = 3). Values in the same column with different letters were significantly different (*p* < 0.05).

**Table 4 foods-14-02163-t004:** Mixolab mixing parameters of composite flour from different highland barley varieties.

Sample	WAC (%)	ST (min)	C2 (Nm)	C3 (Nm)	C5 (Nm)	C3-C4 (Nm)	C5-C4 (Nm)
Chaiqing 1	73.30 ± 0.17 ^g^	9.92 ± 0.02 ^a^	0.49 ± 0.00 ^d^	1.64 ± 0.02 ^b^	3.57 ± 0.00 ^bc^	0.09 ± 0.02 ^fg^	2.19 ± 0.10 ^bcd^
Gankennuo 2	74.10 ± 0.00 ^ef^	9.23 ± 0.16 ^cd^	0.44 ± 0.01 ^e^	1.31 ± 0.00 ^e^	3.68 ± 0.04 ^b^	0.13 ± 0.01 ^d^	2.04 ± 0.03 ^cd^
Beiqing 8	71.30 ± 0.26 ^h^	8.28 ± 0.0.4 ^g^	0.58 ± 0.01 ^a^	1.62 ± 0.01 ^b^	3.81 ± 0.24 ^b^	0.05 ± 0.01 ^h^	2.24 ± 0.22 ^bcd^
Kunlun 14	77.07 ± 0.12 ^a^	9.67 ± 0.02 ^ab^	0.50 ± 0.00 ^d^	1.61 ± 0.01 ^b^	3.46 ± 0.18 ^c^	0.26 ± 0.01 ^a^	1.75 ± 0.03 ^ef^
Kunlun 15	73.87 ± 0.32 ^f^	9.35 ± 0.03 ^cd^	0.51 ± 0.02 ^c^	1.49 ± 0.01 ^d^	3.25 ± 0.01 ^d^	0.16 ± 0.01 ^c^	1.66 ± 0.10 ^fg^
Longzihei	76.50 ± 0.17 ^b^	9.53 ± 0.10 ^bc^	0.51 ± 0.01 ^c^	1.63 ± 0.02 ^b^	2.96 ± 0.01 ^e^	0.10 ± 0.00 ^e^	2.25 ± 0.23 ^bc^
Ximalaya 22	74.27 ± 0.06 ^de^	8.95 ± 0.07 ^ef^	0.42 ± 0.01 ^f^	1.68 ± 0.01 ^a^	2.68 ± 0.03 ^f^	0.06 ± 0.04 ^gh^	1.51 ± 0.02 ^g^
Zangqing 25	75.17 ± 0.29 ^c^	8.85 ± 0.07 ^f^	0.53 ± 0.01 ^b^	1.63 ± 0.00 ^b^	3.47 ± 0.11 ^c^	0.09 ± 0.00 ^ef^	1.97 ± 0.03 ^de^
Zangqing 2000	74.53 ± 0.06 ^d^	9.32 ± 0.37 ^cd^	0.46 ± 0.00 ^e^	1.63 ± 0.01 ^b^	3.38 ± 0.03 ^cd^	0.06 ± 0.01 ^h^	1.96 ± 0.05 ^de^
Zangqing 3000	75.10 ± 0.00 ^c^	9.12 ± 0.01 ^de^	0.48 ± 0.00 ^d^	1.56 ± 0.02 ^c^	4.27 ± 0.15 ^a^	0.18 ± 0.01 ^b^	2.57 ± 0.05 ^a^

Note: WAC, water absorption capacity; ST, stability time; C2, weakening of protein network; C3, rate of starch pasting; C5, starch retrogradation during the cooling period; C3-C4, amylase activity; C5-C4, anti-aging effect of the starch. Results are presented as means ± standard deviations (*n* = 3). Values in the same column with different letters were significantly different (*p* < 0.05).

**Table 5 foods-14-02163-t005:** Cooking parameters of noodles prepared from wheat flour and different highland barley varieties.

Noodles	Optimal Cooking Time (min)	Broken Ratio (%)	Cooking Loss (%)	Cooking Yield (%)
WFNs	3.42 ± 0.01 ^j^	0.00 ± 0.00 ^d^	5.34 ± 0.02 ^f^	143.90 ± 3.41 ^a^
Chaiqing 1	4.08 ± 0.01 ^h^	0.00 ± 0.00 ^d^	5.55 ± 0.25 ^ef^	136.23 ± 6.21 ^b^
Gankennuo 2	4.45 ± 0.01 ^b^	0.00 ± 0.00 ^d^	5.87 ± 0.03 ^cde^	128.80 ± 1.09 ^cde^
Beiqing 8	4.32 ± 0.01 ^d^	5.00 ± 2.36 ^cd^	5.77 ± 0.14 ^ef^	134.59 ± 3.28 ^bc^
Kunlun 14	3.46 ± 0.01 ^i^	11.67 ± 2.35 ^ab^	6.64 ± 0.26 ^a^	119.97 ± 1.60 ^h^
Kunlun 15	4.10 ± 0.02 ^g^	1.67 ± 2.35 ^d^	6.13 ± 0.10 ^bcd^	128.17 ± 2.63 ^def^
Longzihei	4.17 ± 0.01 ^f^	8.33 ± 2.35 ^ab^	6.58 ± 0.45 ^ab^	124.28 ± 0.49 ^efgh^
Ximalaya 22	4.46 ± 0.00 ^b^	0.00 ± 0.00 ^d^	6.31 ± 0.38 ^abc^	133.36 ± 3.73 ^bcd^
Zangqing 25	4.23 ± 0.01 ^e^	5.00 ± 3.36 ^cd^	6.04 ± 0.12 ^cd^	126.68 ± 1.56 ^efg^
Zangqing 2000	4.42 ± 0.01 ^c^	0.00 ± 0.00 ^d^	5.58 ± 0.02 ^ef^	121.27 ± 4.01 ^gh^
Zangqing 3000	5.07 ± 0.02 ^a^	0.00 ± 0.00 ^d^	5.78 ± 0.04 ^def^	134.93 ± 2.56 ^bc^

Note: Results are presented as means ± standard deviations (*n* = 3). Values in the same column with different letters were significantly different (*p* < 0.05).

**Table 6 foods-14-02163-t006:** Color parameters of noodles prepared from wheat flour and different highland barley varieties.

Noodles	*L**	a*	b*
WFNs	93.03 ± 0.37 ^c^	2.03 ± 0.40 ^a^	9.92 ± 0.46 ^a^
Chaiqing 1	85.62 ± 0.47 ^a^	5.45 ± 0.17 ^c^	17.01 ± 0.58 ^bc^
Gankennuo 2	84.30 ± 0.55 ^a^	5.99 ± 0.55 ^bc^	18.16 ± 0.31 ^c^
Beiqing 8	81.04 ± 0.57 ^bc^	6.36 ± 0.28 ^a^	18.13 ± 0.57 ^a^
Kunlun 14	81.83 ± 1.12 ^bc^	6.48 ± 0.32 ^a^	18.56 ± 0.31 ^a^
Kunlun 15	82.95 ± 0.52 ^b^	6.55 ± 0.29 ^a^	16.42 ± 0.19 ^a^
Longzihei	69.04 ± 0.56 ^bc^	6.01 ± 0.48 ^a^	7.92 ± 0.55 ^a^
Ximalaya 22	83.40 ± 0.96 ^a^	6.06 ± 0.57 ^b^	16.41 ± 0.30 ^a^
Zangqing 25	81.60 ± 1.00 ^bc^	6.49 ± 0.28 ^a^	18.09 ± 0.45 ^b^
Zangqing 2000	83.10 ± 0.61 ^bc^	5.68 ± 0.33 a	15.80 ± 0.53 ^a^
Zangqing 3000	82.37 ± 1.15 ^bc^	5.06 ± 0.53 ^a^	15.99 ± 0.49 ^a^

Note: *L**, lightness; a*, redness–greenness; b*, yellowness–blueness. Results are presented as means ± standard deviations (*n* = 3). Values in the same column with different letters were significantly different (*p* < 0.05).

**Table 7 foods-14-02163-t007:** Texture parameters of noodles prepared from wheat flour and different highland barley varieties.

Noodles	Hardness (g)	Springiness (g·s)	Adhesiveness (g·s)	Chewiness (g·s)	Shearing force (g)	Tensile force (g)
WFNs	7399.59 ± 825.52 ^ef^	96.49 ± 0.28 ^b^	−144.65 ± 34.33 ^a^	5448.33 ± 92.67 ^ab^	127.57 ± 3.99 ^e^	15.51 ± 1.76 ^e^
Chaiqing 1	8496.49 ± 117.11 ^ab^	98.84 ± 0.75 ^a^	−191.99 ± 56.27 ^abc^	5543.65 ± 164.17 ^ab^	153.78 ± 4.64 ^bc^	22.52 ± 1.17 ^abc^
Gankennuo 2	7656.46 ± 234.82 ^de^	95.97 ± 0.57 ^bc^	−328.51 ± 14.42 ^e^	4753.36 ± 128.85 ^e^	159.67 ± 9.48 ^ab^	16.77 ± 1.70 ^cd^
Beiqing 8	7711.11 ± 460.95 ^cde^	95.48 ± 0.47 ^c^	−253.63 ± 31.89 ^cde^	5780.07 ± 128.88 ^a^	141.55 ± 2.59 ^d^	20.22 ± 0.79 ^abc^
Kunlun 14	8618.90 ± 251.02 ^a^	96.87 ± 1.21 ^b^	−282.49 ± 63.96 ^de^	4908.08 ± 476.19 ^cde^	164.64 ± 6.08 ^a^	24.01 ± 0.79 ^a^
Kunlun 15	8203.96 ± 782.83 ^abcd^	98.66 ± 0.86 ^a^	−278.34 ± 78.95 ^de^	5288.93 ± 492.55 ^bc^	124.57 ± 3.18 ^e^	15.62 ± 1.50 ^e^
Longzihei	8597.02 ± 290.16 ^a^	94.94 ± 0.47 ^c^	−235.10 ± 57.90 ^cd^	4805.59 ± 434.00 ^de^	156.01 ± 6.95 ^abc^	16.69 ± 1.41 ^cd^
Ximalaya 22	8413.05 ± 320.35 ^abc^	98.25 ± 0.72 ^a^	−148.26 ± 22.78 ^ab^	5242.82 ± 102.89 ^bcd^	152.18 ± 9.80 ^bc^	23.29 ± 0.55 ^bc^
Zangqing 25	8488.68 ± 167.90 ^ab^	98.38 ± 0.65 ^a^	−226.95 ± 67.43 ^abcd^	5446.11 ± 271.43 ^ab^	148.26 ± 5.21 ^cd^	20.32 ± 1.09 ^abc^
Zangqing 2000	8090.20 ± 232.93 ^abcde^	98.42 ± 0.28 ^a^	−213.13 ± 49.02 ^abcd^	4620.93 ± 255.20 ^e^	140.18 ± 9.32 ^d^	18.19 ± 0.71 ^cd^
Zangqing 3000	7850.20 ± 593.81 ^bcde^	96.94 ± 0.48 ^b^	−227.18 ± 18.56 ^bcd^	5769.82 ± 240.43 ^a^	142.27 ± 4.19 ^d^	19.38 ± 0.71 ^bcd^

Note: Results are presented as means ± standard deviations (*n* = 3). Values in the same column with different letters were significantly different (*p* < 0.05).

**Table 8 foods-14-02163-t008:** In vitro starch digestion kinetics and eGI.

Noodles	*K* × 10^−2^	*C* * _∞_ *	eGI
WFN	3.06 ± 0.15 ^a^	82.41 ± 1.35 ^a^	89.97 ± 0.21 ^a^
Chaiqing 1	1.73 ± 0.12 ^d^	59.39 ± 1.72 ^g^	65.78 ± 0.09 ^ef^
Gankennuo 2	1.45 ± 0.13 ^g^	80.23 ± 3.34 ^b^	78.24 ± 0.02 ^b^
Beiqing 8	1.23 ± 0.13 ^i^	78.62 ± 3.99 ^c^	75.68 ± 0.23 ^c^
Kunlun 14	1.60 ± 0.07 ^f^	48.96 ± 1.01 ^k^	65.04 ± 0.08 ^f^
Kunlun 15	1.06 ± 0.04 ^j^	64.12 ± 1.39 ^e^	70.42 ± 0.18 ^d^
Longzihei	2.02 ± 0.11 ^b^	50.71 ± 0.97 ^j^	65.06 ± 0.19 ^f^
Ximalaya 22	1.34 ± 0.07 ^h^	54.78 ± 1.30 ^h^	66.05 ± 0.69 ^e^
Zangqing 25	1.67 ± 0.12 ^e^	60.39 ± 1.88 ^f^	74.57 ± 0.23 ^c^
Zangqing 2000	1.48 ± 0.05 ^g^	53.84 ± 0.86 ^i^	66.35 ± 0.05 ^e^
Zangqing 3000	1.84 ± 0.14 ^c^	68.24 ± 2.05 ^d^	70.30 ± 0.91 ^d^

Note: Results are presented as means ± standard deviations (*n* = 3). Values in the same column with different letters were significantly different (*p* < 0.05).

## Data Availability

The original contributions presented in the study are included in the article, further inquiries can be directed to the corresponding authors.

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
