# Peer review of "Effects of Different Highland Barley Varieties on Quality and Digestibility of Noodles"

_foods, 2025, doi:10.3390/foods14132163_

Round 1
Reviewer 1 Report
Comments and Suggestions for Authors
Thank you for the opportunity to review your manuscript entitled “Effects of Different Highland Barley Varieties on Noodle Quality and Digestibility.” The study addresses a relevant topic in cereal technology and nutrition and provides important findings on the selection of barley varieties for the development of functional noodles. However, several points require review or clarification before publication; comments were added. See atachment.

Author Response
Dear editors,
Thanks for your letter informing me that my manuscript (foods-3662356) has been reviewed and requested revisions. I carefully read the comments of the review and revised my manuscript as requested. The changes I made to the original manuscript are highlighted with red text under the revised model. In addition, the full revision can be found in the attachment below.
Thank you for your hard work. If you have any questions, we would be very grateful for more information. Looking forward to hearing from you.
- Consider explaining how Highland barley compares to other functional grains in the noodle recipe.
Re: Thanks for the valuable advice. The main advantage of Highland barley in noodle recipes is its high β-glucan content, which effectively lowers GI by slowing down starch hydrolysis. In addition, highland barley contains high levels of dietary fiber compared to other functional grains such as buckwheat and quinoa. P2, lines 51-54.
- The rationale for choosing ten specific barley varieties may be more specific: for example, they are the most widely marketed, they differ in their genetic composition and, therefore, in their chemical composition.
Re: Thank you for your question. We systematically analyzed 10 varieties of plateau barley (Hordeum vulgare L.) from major producing areas in China. These varieties are selected and widely cultivated for their representativeness. P2, lines 81-83.
- How did they arrive at this ratio? Literature? Initial Discussion? It's not clear.
Replies: Thank you for your constructive comments. The ratio of the noodle recipe was determined based on the results of preliminary experiments. P4, lines 114-116.
- [27] Lack of references.
Re: Refer to the following: P4, line 142.
Li M, Zhu K X, Sun Q J, et al.Quality characteristics, structural changes and storage stability of semi-dry noodles induced by moderate dehydration——Understanding the quality changes of semi-dry noodles[J].Food Chemistry, 2016, 194: 797-804.https://doi.org/10.1016/j.foodchem.2015.08.079
5 [55] Lack of references.
Re: Refer to the following: P12, line 320.
Effect of Flour Particle Size on Quality Characteristics of Recombinant Whole Wheat Flour and Southern Manchu[J]. LWT-Food Science and Technology, 2017, 82: 147-153.https://doi.org/10.1016/j.lwt.2017.04.025
- More specifically, is it the weight after boiling?
Re: Thanks for the valuable advice. Cooking yield is measured by the following formula: (the corresponding reference material is inserted in the text). P5, lines 158-161.
Cooking rate (%) = (m2-m 1)/m1×100
where m2 represents the weight of the wet noodles after boiling and m1 represents the mass of the dry noodles before boiling.
- Can you explain in more detail the relationship between C∞ and eGI?
Re: Thanks for the valuable advice. Parameter C∞ is derived by fitting in vitro digestion data to a first-order kinetic model. Studies have shown that eGI is affected by C∞ and the kinetic constant k. P20, lines 501-503.
- What does the best quality compare to? It can be helpful to specify features.
Re: Chai Ching No. 1 has the best quality in three areas: (1) product performance (better cooking and texture quality), (2) processing adaptability (longer settling time and ideal C2 value), and (3) nutritional quality (high protein and high β-glucan content). As a result, it is considered the perfect ingredient for making high-quality barley noodles. Page 23, lines 597-602.
Sincerely.
Sincerely
Wang Lili
8 June, 2025
Email: wlland2013@163.com

Reviewer 2 Report
Comments and Suggestions for Authors
This study investigated the physicochemical characteristics of different barley cultivars. The experiments were conducted clearly and appropriately in alignment with the research objectives. However, the discussion of the results is insufficient. Furthermore, the manuscript includes a number of subjective interpretations rather than maintaining a strictly objective presentation of the data, which necessitates substantial revisions. A more thorough and critical discussion of the findings should be added to enhance the scientific value of the research.
1. Lines 55–56: The assertion that the low glycemic index (GI) is solely due to slow carbohydrate digestibility is an oversimplification. While reduced digestibility can contribute to a lower GI, other factors such as the presence of viscous dietary fibers, resistant starch, food matrix structure, and the interaction between macronutrients also play significant roles. A more comprehensive explanation of the mechanisms underlying the low GI observed in the study is necessary, supported by relevant literature.
2. Lines 102–103: The rationale for selecting the specific formulation ratio of 50:42.5:7.5 is not provided. A clear explanation should be included to justify the basis for this composition, whether it is based on preliminary experiments, functional properties, or previous literature.
3. Lines 175–177: Please provide appropriate references for Equations (4), (5), and (6). The source or derivation of these equations should be clearly cited to ensure transparency and reproducibility.
4. Lines 184–206: A detailed explanation is required regarding how the samples not listed in Table 1 were obtained.
5. Lines 184–206: The results of each experiment should be compared with findings reported in previous studies.
6. Lines 214–228: Amylose content is also associated with final viscosity. Please explain why the final viscosity of Kunlun 14 is lower than that of Gankennuo 2, considering their respective amylose contents and other possible contributing factors.
7. Lines 224–248: Although Longzihei and Beiqing 8 have similar amylose contents, there is a noticeable difference in their peak gelatinization temperatures. Please provide an explanation for this discrepancy.
8. Lines 224–248: The relationship between amylose content and peak gelatinization temperature does not appear to hold for Longzihei and Xymalaya 22. An explanation is needed to account for this deviation.
9. Lines 251–297: The discussion currently focuses only on a few selected correlations, leaving out a broader interpretation of the findings. A more thorough and integrated discussion of the overall results is needed to strengthen the scientific value and coherence of the study.
10. Lines 313–316: It is stated that dense granules result in less stickiness and firmness, as well as increased cooking loss. However, the specific results being referenced are unclear. Please provide the corresponding data and clearly explain which experimental findings support this statement.
11. Lines 344–361: The criteria used to determine the optimal cooking time are not clearly described. Please provide a detailed explanation of how the optimal cooking time was defined or measured in this study.
12. Lines 428–432: Please explain why the Kunlun 14 sample exhibited a higher level of resistant starch (RS).
13. Lines 405–453: The differences in RDS, SDS, and RS contents among the samples should be explained in relation to the data presented in Table 2. A clear discussion linking these starch fractions to the physicochemical properties of each sample would enhance the interpretation of the results.
Author Response
Dear editor,
Thanks for your letter to inform that my manuscript (foods-3662356) has been reviewed and asked to be revised. I have read the comments of the reviews carefully and revised my manuscript as required. The changes that I’ve made to the original manuscript are highlighted by using red colored text under the revised model. In addition, the answers point by point to the questions that were raised by all reviewers are displayed in the attachment.
Thanks for your hard work. If any questions, we will be very grateful for your more information. Looking forward to receiving your letter.
1. Lines 55–56: The assertion that the low glycemic index (GI) is solely due to slow carbohydrate digestibility is an oversimplification. While reduced digestibility can contribute to a lower GI, other factors such as the presence of viscous dietary fibers, resistant starch, food matrix structure, and the interaction between macronutrients also play significant roles. A more comprehensive explanation of the mechanisms underlying the low GI observed in the study is necessary, supported by relevant literature.
Reply: Thank you for your constructive suggestions. We have already explained the mechanisms underlying the low GI in the text. The corresponding sentences have been added and can be found in the revised version P2, Lines 62-67. The influence of B-glucan has also been explained in the text. P2, Lines 75-77.
2. Lines 102–103: The rationale for selecting the specific formulation ratio of 50:42.5:7.5 is not provided. A clear explanation should be included to justify the basis for this composition, whether it is based on preliminary experiments, functional properties, or previous literature.
Reply: Thanks for your reminder. The proportion of the noodle recipe was determined based on the results of preliminary experiments. P4, Lines 114-116.
3. Lines 175–177: Please provide appropriate references for Equations (5), Equations (6) and Equations (7). The source or derivation of these equations should be clearly cited to ensure transparency and reproducibility.
Reply: Thank you for your constructive suggestions. The corresponding references have now been inserted in the text. P5, Lines 188-190.
4 Lines 184–206: A detailed explanation is required regarding how the samples not listed in Table 1 were obtained.
Reply:Thank you for your reminder. All ten highland barley varieties used in our experiment were provided by the Qinghai Academy of Agriculture and Forestry Sciences and have all been listed in table 1. P3, Line 97-98.
5 Lines 184–206: The results of each experiment should be compared with findings reported in previous studies.
Reply: Thank you for your constructive suggestions. These results have been compared with previous studies, which were added in P6, Lines 208-211, P6, Lines 224-225, P6 , Lines 201-204, P14, Lines 342-343, P17, Lines 420-422, P21, Lines 522-524, 532-535, 537-540, 546-548.
6 Lines 214–228: Amylose content is also associated with final viscosity. Please explain why the final viscosity of Kunlun 14 is lower than that of Gankennuo 2, considering their respective amylose contents and other possible contributing factors.
Reply: Thank you for your question. The reduced final viscosity observed in Kunlun 14 compared to Gankennuo 2 was attributable not only to differences in amylose content, but also to variations in total starch content, amylopectin chain-length distribution, and interactions with non-starch components. P9 , Lines 249-252.
7 Lines 224–248: Although Longzihei and Beiqing 8 have similar amylose contents, there is a noticeable difference in their peak gelatinization temperatures. Please provide an explanation for this discrepancy.
Reply: Thank you for your question. Although the amylose content of Longzihei and Beiqing 8 was similar, there was a significant difference in their peak temperatures. This is because the peak temperature is not only affected by the content of amylose, but also by the total starch content, other components such as proteins and lipids, and the distribution of amylopectin chain length. In future studies, we will further investigate the effect of these factors collectively influence on the peak gelatinization temperature of starch. P9-P10, Lines 271-275.
8 Lines 224–248: The relationship between amylose content and peak temperature does not appear to hold for Longzihei and Ximalaya 22. An explanation is needed to account for this deviation.
Reply: Thank you for your question. The relationship between amylose content and peak temperature does not appear to hold for Longzihei and Ximalaya 22. This is because the peak temperature of starch in the samples is influenced not only by amylose content but also by other critical factors, such as the molecular structure of starch granules, crystallinity characteristics, and interactions with non-starch components. P10, Lines 276-280.
9 Lines 251–297: The discussion currently focuses only on a few selected correlations, leaving out a broader interpretation of the findings. A more thorough and integrated discussion of the overall results is needed to strengthen the scientific value and coherence of the study.
Reply: Thank you for your constructive suggestions. We have conducted a more comprehensive and integrated analysis, which was added to P21, Lines 522-527, 532-535, 537-540, 546-548.
10 Lines 313–316: It is stated that dense granules result in less stickiness and firmness, as well as increased cooking loss. However, the specific results being referenced are unclear. Please provide the corresponding data and clearly explain which experimental findings support this statement.
Reply: Thank you for your suggestion. We have supplemented the relevant literature. Please refer to the revised version for details. P14, Lines 350-356.
11 Lines 344–361: The criteria used to determine the optimal cooking time are not clearly described. Please provide a detailed explanation of how the optimal cooking time was defined or measured in this study.
Reply: Thank you for your question. During cooking, the optimal cooking time of noodles is typically identified when the starches were completely gelatinized in the noodles (indicated by the disappearance of the white starch granules The time at which this occurs is considered the optimal cooking time. This criterion has been clearly defined in P4, Lines 151-154.
12 Lines 428–432: Please explain why the Kunlun 14 sample exhibited a higher level of resistant starch (RS).
Reply: Thank you for your question. RS exhibited significantly higher levels in Kunlun 14, which might be Kunlun 14 exhibited large cracks and its stomata were irregular (Figure 1D), which was unconducive to the entry of enzymes into the sample. In addition, the dietary fiber content of Kunlun 14 was 27.98%. Dietary fiber can form a physical barrier, hindering the hydrolysis of starch by digestive enzymes, while increasing the viscosity of intestinal contents and delaying the digestion process of starch. P19, Lines 472-478.
13 Lines 405–453: The differences in RDS, SDS, and RS contents among the samples should be explained in relation to the data presented in Table 2. A clear discussion linking these starch fractions to the physicochemical properties of each sample would enhance the interpretation of the results.
Reply: Thank you for your constructive suggestions. We respectively explained how the relevant factors affected the physicochemical properties of the samples. The specific content can be found in the revised version. P22, Lines 556-562. P19, Lines 478-487, Lines 491-495.
Best regards.
Yours sincerely,
Lili Wang
8 June, 2025
E-mail: wlland2013@163.com

Round 2
Reviewer 2 Report
Comments and Suggestions for Authors
The response to the previous review has been adequately addressed. The manuscript appears to provide sufficient discussion of the research findings.
Author Response
Dear editor,
Thanks for your letter to inform that my manuscript (foods-3662356) has been reviewed and asked to be revised. I have read the comments of the reviews carefully and revised my manuscript as required. The changes that I’ve made to the original manuscript are highlighted by using red colored text under the revised model. In addition, the answers point by point to the questions that were raised by all reviewers are displayed in the attachment.
Thanks for your hard work. If any questions, we will be very grateful for your more information. Looking forward to receiving your letter.
- Line 353. Sentence must be reviewed.
Reply: Thank you for your valuable suggestion. This phenomenon aligns with the findings of Wang et al. who demonstrated that the structural collapse of starch granules, characterized by the melting of the crystalline zone, the dissociation of double helices, and the breakage of hydrogen bonds, is a critical step in gelatinization. P14, Lines 350-353.
- Line 473. What did the authors mean by "stomata"? Please revise the sentence as a whole as it sounds confuse.
Reply: Thank you for your valuable suggestion. RS exhibited significantly higher levels in Kunlun 14, which might be Kunlun 14 exhibited large cracks and irregular surface pores (Fig. 1D), which hindered enzyme penetration into the sample. P19, Lines 472-474.
Best regards,
Lili Wang
15 June, 2025
E-mail: wlland2013@163.com
